# Instruction Tuning Large Language Models to Understand Electronic Health Records

**Zhenbang Wu**[1,2,3], **Anant Dadu**[2,3], **Mike Nalls**[2,3], **Faraz Faghri**[2,3*], **Jimeng Sun**[1*]
[1]UIUC,  [2]National Institutes of Health,  [3]Data Tecnica
[*]Co-corresponding authors

## Abstract

Large language models (LLMs) have shown impressive capabilities in solving a wide range of tasks based on human instructions. However, developing a conversational AI assistant for electronic health record (EHR) data remains challenging due to (1) the lack of large-scale instruction-following datasets and (2) the limitations of existing model architectures in handling complex and heterogeneous EHR data. In this paper, we introduce `MIMIC-Instr`, a dataset comprising over 400K open-ended instruction-following examples derived from the MIMIC-IV EHR database. This dataset covers various topics and is suitable for instruction-tuning general-purpose LLMs for diverse clinical use cases. Additionally, we propose `Llemr`, a general framework that enables LLMs to process and interpret EHRs with complex data structures. `Llemr` demonstrates competitive performance in answering a wide range of patient-related questions based on EHR data. Furthermore, our evaluations on clinical predictive modeling benchmarks reveal that the fine-tuned `Llemr` achieves performance comparable to state-of-the-art (SOTA) baselines using curated features. The dataset and code are available at `https://github.com/zzachw/llemr`.

## 1  Introduction

EHRs document a patient's medical history and care, including demographics, diagnoses, laboratory test results, medication prescriptions, and clinical notes [Evans, 2018]. Despite the potential benefits in supporting clinical decision-making and care coordination, EHR systems also lead to physician burnout due to challenges in navigating the user interface, the large volume of data that needs to be reviewed for each medical decision, and the extra clerical tasks directed to physicians [Melnick et al., 2020, Tajirian et al., 2020, DeChant et al., 2019]. Previous studies show that physicians spend an average of 3.17 hours daily on EHR systems [Overhage and McCallie, 2020]. This not only detracts from patient care but also reduces the time physicians can spend interacting directly with patients.

Advances in LLMs offer an opportunity to streamline EHR processes and ease the load on healthcare providers. LLMs have revolutionized natural language processing fields in tasks such as question answering [Wei et al., 2022a, Touvron et al., 2023], visual understanding [Liu et al., 2023, Li et al., 2023a, 2022, 2023b], reasoning [Wei et al., 2022b, Brown et al., 2020], and code generation [Black et al., 2022]. They have demonstrated remarkable capabilities in understanding complex inputs and following human instructions to solve diverse tasks. Recent works have further shown that LLMs can achieve expert-level performance on multiple-choice questions from medical licensing exams [Singhal et al., 2022, 2023]. However, despite these advances, developing a conversational AI assistant specifically for EHR data remains a significant challenge.

**Challenge 1: Lack of large-scale instruction-following data.** LLMs are typically fine-tuned on large-scale instruction-following datasets to understand user instructions and perform a variety of tasks [Wei et al., 2022a]. These datasets are created using manually defined templates or with

38th Conference on Neural Information Processing Systems (NeurIPS 2024) Track on Datasets and Benchmarks.

the assistance of LLMs. The construction process requires substantial efforts and becomes even more complex when data must be paired with patient EHRs. Thus, most prior works mainly focus on the clinical notes [Kweon et al., 2024, Lehman et al., 2022, Yue et al., 2021], as generating instruction-following data from free text is comparatively straightforward. However, a substantial amount of information exists solely within structured EHR data (e.g., relational tables). Although some question-answering (QA) datasets are based on structured EHR data [Pampari et al., 2018, Lee et al., 2023, Tang et al., 2023], they mainly focus on factoid extraction and lack alignment with real-world clinical decision-making, which often requires complex reasoning. Moreover, existing datasets are limited in size [Fleming et al., 2024], ranging from thousands to tens of thousands of examples, which is insufficient for effective LLM instruction tuning.

**Challenge 2: Limitations of existing model architectures in handling complex and heterogeneous EHR data.** Due to the complex schemas and various standardizations of EHR data [Gamal et al., 2021], most existing clinical predictive models depend heavily on manual data preprocessing [Harutyunyan et al., 2019, Choi et al., 2016a,b]. This preprocessing includes steps like feature selection, code mapping, unit standardization, value normalization, and imputation. Such manual processes demand significant time and expertise and may introduce human errors and biases into the data. While recent works have attempted to develop a unified foundation model for EHRs [Hur et al., 2024, Kim et al., 2024], these models are generally limited to specific tasks and lack interactive capabilities.

In this paper, we introduce `MIMIC-Instr`, a dataset of over 400K EHR-grounded instruction-following examples based on the publicly available MIMIC-IV EHR database [Johnson et al., 2023]. This dataset is divided into two parts: **(1) Schema alignment subset:** A set of 350K QA pairs was constructed from over 100 templates and subsequently paraphrased using GPT-3.5 [1]. These questions query various information from the structured EHR data, such as patient demographics, diagnoses, treatment histories, and test results. They are designed to train LLMs on the ability to navigate and extract specific information from the complex and heterogeneous EHR data. **(2) Clinical reasoning subset:** Another set of 50K QA pairs was generated from discharge summaries with GPT-3.5. Discharge summaries capture the complexities of patient cases and the rationales behind medical decisions. This subset challenges LLMs to go beyond simple fact extraction, engaging in deeper clinical reasoning tasks such as understanding the progression of a patient's condition, predicting possible complications, and suggesting appropriate follow-up actions.

To address the second challenge, we propose `Llemr`, an instruction-tuned LLM for electronic medical records (EMRs) [2]. We adopt the Medical Event Data Standard [Arnrich et al.] and represent each patient's EHR data as a stream of clinical events (e.g., procedures, prescriptions, and transfers). Each event is formatted as a triplet of timestamp, type, and value, such as (2024-03-11 13:32:26, Lab, Hemoglobin 12 g/dl), which can be combined into sentence-like representations. This format simplifies inputs and is robust to variations in EHR schemas and standards. While the concatenated event sequence can be directly input into LLMs, it often exceeds the context length of LLMs for patients with extensive event histories. Inspired by REMed [Kim et al., 2024], we utilize ClinicalBERT [Alsentzer et al., 2019] to first encode each event into an embedding, and then feed the sequence of event embeddings into the LLM with an additional mapping layer (i.e., a linear projection). This approach largely reduces the input length and speeds up the training. `Llemr` is trained in a curriculum learning fashion: it initially learns to navigate through EHR data via 350K schema alignment examples, and then progresses to more complex reasoning with the 50K clinical reasoning examples.

`Llemr` exhibits excellent ability to answer diverse inquiries about a patient. Further, our evaluation on standard clinical predictive benchmarks shows that the fine-tuned `Llemr` achieves performance comparable to SOTA baselines using manually curated features.

In summary, this paper makes the following contributions:

- **Clinical instruction-following data.** We created a dataset of 400K instruction-following examples based on the MIMIC-IV database. This dataset enables instruction-tuning of general LLMs to better understand EHRs.

---

[1] We used Azure's HIPAA-compliant platform in accordance with PhysioNet's regulations.
[2] We use EMR and EHR interchangeably in this paper.

Table 1: Comparison between `MIMIC-Instr` and existing clinical question-answering / instruction-following datasets.

| Dataset | Size | Source | Format | Answer Type |
|---|---|---|---|---|
| MedQA [Jin et al., 2020] | 13K | US medical licensing exam | Question + Answer | Multi Choice |
| MedMCQA [Pal et al., 2022] | 6K | AIIMS and NEET PG entrance exams | Question + Answer | Multi Choice |
| PubMedQA [Jin et al., 2019] | 0.5K | PubMed literature | Question + Context + Answer | Multi Choice |
| MMLU clinical [Hendrycks et al., 2021] | 1K | US Medical Licensing Examination | Question + Answer | Multi Choice |
| EHRSQL [Lee et al., 2023] | 24K | MIMIC-III | Question + Answer | SQL |
| EHRNoteQA [Kweon et al., 2024] | 0.9K | MIMIC-IV | Question + Note + Answer | Free Text |
| MedAlign [Fleming et al., 2024] | 0.9K | EHRs (Stanford University) | Question + EHR + Answer | Free Text |
| `MIMIC-Instr` | 400K | MIMIC-IV | Question + EHR + Answer | Free Text |

- **Foundation model for EHR.** We introduce `Llemr`, a general framework to empower LLMs to perform both information extraction and clinical reasoning on EHR data.

- **Open-source.** We will release the instruction-following data via PhysioNet [3], and also share the code and model weights to facilitate future research.

# 2 Related Work

## 2.1 Clinical Instruction-Tuning Data

Instruction tuning fine-tunes a pre-trained LLM using pairs of instructions and responses. It generalizes LLMs' capabilities beyond next token prediction to diverse new tasks described with instructions. Generally, the instruction tuning datasets are constructed either with manually-defined templates or LLMs like GPT. While there are increasing interests in adapting LLMs to the clinical domain, existing works mainly focus on broad clinical tasks with natural language inputs, such as answering medical licensing exam questions and consumer queries [Singhal et al., 2022, Johri et al., Tu et al., 2024, 2023], information extraction and text summarizing [Tran et al., 2023, Zhang et al., 2024], and ICD coding [Wang et al., 2024]. These tasks are typically based on natural language as inputs and thus the instruction tuning data is relatively easy to generate. However, existing works fall short in instruction tuning a LLM to understand EHR data. While there are some recent benchmarks evaluating the capability of LLM on EHR data, they either focus on information extraction tasks [Lee et al., 2023, Shi et al., 2024], or solely perform model evaluation [Kweon et al., 2024, Fleming et al., 2024, Zakka et al., 2024], as the datasets are too small to enable instruction tuning. To bridge this gap, we release a dataset of 400K instruction-response examples on patient EHR data covering a broad range of topics and can be used to instruction tune general-purpose LLMs to understand EHR data.

## 2.2 Foundation Model for EHR

With the wide adoption of EHR systems, there has been growing interest in utilizing deep learning models in interpreting and analyzing EHR data to assist clinical decision-making and improve patient outcome [Choi et al., 2016b, Tan et al., 2022, Choi et al., 2017, Zhang et al., 2021, Li et al., 2020, Xu et al., 2024]. However, real-world EHR data is often very messy and has complex schemas. Thus, most existing methods typically require manual feature selection and complex data preprocessing to harmonize the data. This process requires domain expertise and is very time-consuming. Yet, the developed model is often task-specific and the same process needs to be performed again for new tasks. In recent years, language models have demonstrated remarkable capability in understanding diverse text inputs [Brown et al., 2020]. Thus, some recent methods try to utilize them to encode clinical events and eliminate the need for feature selection and data preprocessing [Hur et al., 2024, Kim et al., 2024]. However, they still follow the task-specific supervised training paradigm and do not fully utilize the power of LLMs. In this work, we also follow recent trends in converting clinical events into text and utilize LLMs to interpret them. But we take a step further in tuning LLMs to follow instructions and generalize to unseen tasks.

---

[3] Under the PhysioNet Credentialed Health Data License: `https://www.physionet.org/content/ehr-ds-qa/view-license/1.0.0/`.

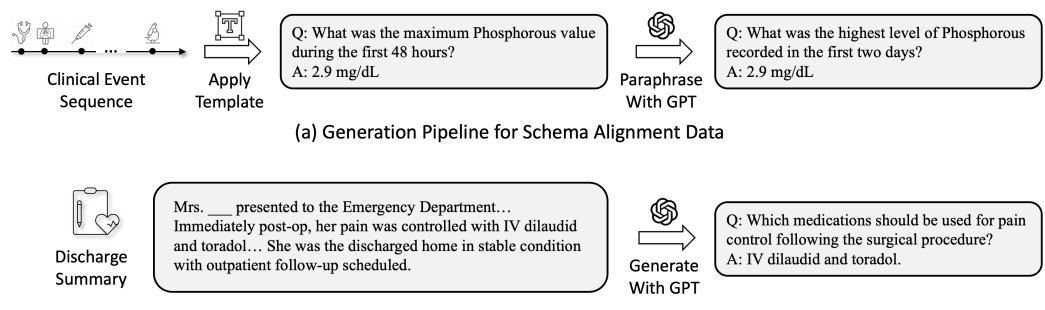

(a) Generation Pipeline for Schema Alignment Data

(b) Generation Pipeline for Instruction Following Data

Figure 1: Illustration of the construction process of `MIMIC-Instr`, a dataset of 400K EHR-grounded instruction-following examples based on the publicly available MIMIC-IV database. It serves two sequential purposes: adapting LLMs to the EHR schema and teaching LLMs to perform in-depth clinical reasoning.

## 3 Preliminaries

EHR records comprehensive clinical information about a patient, including demographics, diagnoses, prescriptions, laboratory and microbiology tests, vital signs, and more. Most existing clinical predictive models rely on expert-defined features to construct a predictive pipeline [Harutyunyan et al., 2019]. This process is labor-intensive, requiring significant domain expertise, and is often repetitive across different tasks. To alleviate this, we adopt the Medical Event Data Standard [Arnrich et al.] and represent each patient's EHR data as a sequence of events. Each event consists of a timestamp (e.g., 2024-03-11 13:32:26) denoting when the event occurred, a type (e.g., Lab) denoting the category of the event, and a value (e.g., Hemoglobin 12 g/dl) denoting the content of the event. This approach allows a unified representation across different EHR schemas.

Besides clinical event sequence, EHR data often includes] discharge summaries written by doctors or nurses at the time of discharge. Discharge summaries provide a comprehensive overview of a patient's hospital stay, detailing the reason for admission, treatments provided, patient's responses to treatment, and recommendations for follow-up care. This data is usually not used in clinical predictive modeling as it is only available at hospital discharge. In this work, we only leverage discharge summaries as a complementary source to generate instruction-following data.

## 4 Clinical Instruction-Following Data

With the growing digitization of healthcare, EHR data is now routinely collected [Evans, 2018]. However, clinical instruction-following data remains limited because its creation is time-consuming and requires significant domain expertise. Therefore, many existing works resort to medical exam questions to tune and evaluate LLM performance in the medical domain [Singhal et al., 2023, Tu et al., 2023]. Yet, these exam-style questions are quite different from how doctors interact with LLMs in real-world clinical practice. To adapt general-purpose LLMs for clinical use, we must first enable them to understand EHR schemas and to reason effectively over EHR data. Inspired by the success of recent works in utilizing GPT to generate instruction-following data [Liu et al., 2023, Li et al., 2023a], we created a clinical instruction-following dataset through a machine-human co-curation process. This dataset includes two subsets: information extraction data and clinical reasoning data, which are used at different training stages. An overview of the data generation process can be found in Figure 1.

### 4.1 MIMIC-IV EHR Database Preparation

We construct our cohort from ICU patients in the MIMIC-IV [Johnson et al., 2023] database. This database contains 51K patients admitted to the ICU at Beth Israel Dea-

Table 2: Example questions from `MIMIC-Instr`.

| | |
|---|---|
| **Schema Alignment** | What was the ethnicity of the individual? |
| | What was the patient's Cr level at the time of discharge? |
| | What was the highest recorded Blood Oxygen Blood Gas level on day 13? |
| | Which organisms were detected in the MRSA SCREEN sample after 104.00 hours? |
| | What was the average measurement of Blood Lymphocytes Hematology within the initial 12-hour period? |
| **Clinical Reasoning** | What was the reason for the patient's hypotension upon presentation to the emergency department? |
| | What interventions were performed for the patient's pancreatic pseudocyst? |
| | What is the recommended follow-up plan for the patient's abdominal pain and gastrointestinal symptoms? |
| | Why was a statin not started for the patient despite other medications being titrated to effect? |
| | What is the recommended dose for levalbuterol for the patient's severe COPD exacerbation? |

coness Medical Center. We filter out patients without discharge summaries, with more than two ICU stays per hospital admission, and with negative ICU or hospital length-of-stay. We then select the following tables from MIMIC-IV: hosp/patients, hosp/admissions, hosp/diagnosis, hosp/labevents, hosp/microbiologyevents, hosp/prescriptions, hosp/transfers, icu/icustays, icu/inputevents, icu/outputevents, icu/procedureevents. Note that icu/chartevents table is excluded due to two reasons: it contains dense bedside monitor data, which is better treated as a time series rather than as an event sequence, and it has substantial overlap with other tables, such as hosp/labevents [Johnson et al., 2023]. In the end, we have a total of 55846 admissions. We hold out 10% each for validation and testing.

## 4.2 Data for Schema Alignment

As described in Section 3, EHR data has a schema that is fundamentally different from general text. To bridge this gap, we first created a set of 350K instruction-tuning examples focused on clinical information extraction. Specifically, for each type of clinical event, we developed a set of question templates (e.g., "which {measurement_name} performed on the {specimen_name} were abnormal {time_period}?"). These templates query diverse information from patient EHR data in the MIMIC-IV database. Each question template is paired with a manually crafted Python script that extracts the ground-truth answer from the corresponding EHR table.

Given a patient's EHR data, we randomly select a template to generate a corresponding question-answer pair (e.g., Q: "Which Blood Gas measurement on the Blood specimen were abnormal at the 650.05 hour?" A: "Calculated Total CO2, pCO2, pO2."). Since the generated QA pairs all follow some fixed template, which limits their effectiveness for training LLMs to interpret diverse instructions, we leveraged GPT-3.5 to paraphrase the generated QA pairs without altering their meanings (e.g., Q: "Show me the abnormal blood gas measurements at the 650.05 hours?" A: "The calculated total CO2, pCO2, pO2 were abnormal.")

In this way, we generated 350K QA pairs focused on information retrieval. This set of instruction-tuning QA pairs primarily asks about the extraction and aggregation of specific factual information from EHR data, serving as a foundational step for enabling LLMs to perform deeper clinical reasoning on EHR data.

## 4.3 Data for Clinical Reasoning

Expectations for clinical assistant AI often go beyond information extraction to following various instructions and performing clinical reasoning. To align the model with this goal, we created diverse instruction-following data focused on clinical reasoning using GPT-3.5. Specifically, we prompted GPT-3.5 to generate questions and answers that resemble those doctors might ask in real-world clinical settings. We also manually created few-shot examples in the prompt to demonstrate how to generate high-quality QA pairs.

However, struggled to interpret raw clinical event sequences (converted to text) due to the unique structure of EHR data. So instead, we leveraged complimentary discharge summaries from the MIMIC-IV database as input to generate QA pairs. Compared to raw clinical event sequences, discharge summaries provide a more concise overview of the patient's hospital trajectory and often

include the rationale behind treatments and plans for future care. This makes the QA data generated from discharge summaries better suited for clinical reasoning tasks. In this way, we generated another 50K QA pairs to equip the model with clinical reasoning abilities.

Note that since the discharge summaries are usually generated at the end of hospital admission and are not available for many real-time clinical predictive tasks, we only use discharge summaries to generate instruction-tuning data. The inputs to our foundation clinical model (introduced next) consist solely of clinical event sequences.

## 5 Llemr: A Foundation Model for EHR Data

With the generated data, we further propose Llemr, a simple yet effective model for EHR data.

### 5.1 Architecture

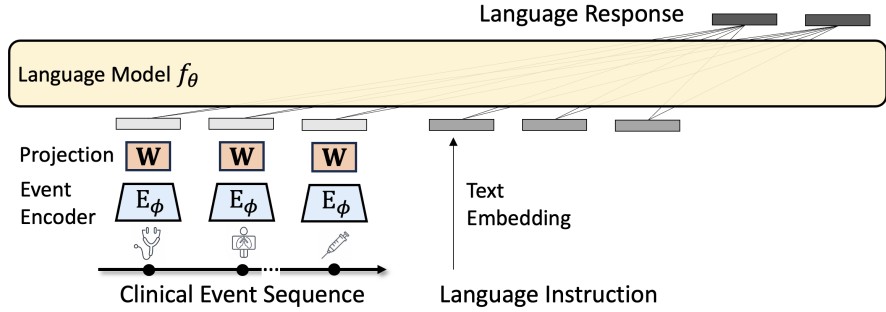

Figure 2: Model architecture of Llemr.

Given a patient's EHR event sequence and a language instruction from the user, Llemr learns to generate a free-text response. Due to the high volume of events associated with MIMIC-IV ICU patients, instead of directly encoding raw events, Llemr utilizes an additional event encoder. Specifically, for each patient's event sequence, ClinicalBERT [Alsentzer et al., 2019] is first used to encode each event into an embedding. Then, Llemr applies a linear projection layer to map the event embeddings to the word embedding space. The sequence of event embeddings is further concatenated with the token embeddings for the language instruction and fed into the backbone LLM (i.e., vicuna-7b-v1.5 [Chiang et al., 2023]).

### 5.2 Training with Curriculum Learning

Inspired by LLaVA [Liu et al., 2023], we adopt a two-stage curriculum training procedure to first bridge the schema gap between general text and EHR data, and then equip the model with clinical reasoning abilities.

#### 5.2.1 Stage 1: Training for Schema Alignment

We utilize the 350K QA pairs generated from templates and paraphrased by GPT-3.5. For each patient, given the event sequence and language instruction, we ask the model to generate the corresponding response. We keep both the event encoder and LLM backbone frozen and only update the projection matrix. In this way, we can teach LLMs to interpret the outputs from the event encoder and bridge the schema gap between general text and EHR data.

#### 5.2.2 Stage 2: Training for Clinical Reasoning

In this stage, we continue to tune the model to follow more complex instructions and perform clinical reasoning. We utilize the other 50K QA pairs for clinical reasoning and tune both the LLM and projection matrix. The weights of the event encoder are kept frozen. This allows Llemr to go beyond

Table 3: Evaluation on the performance of `Llemr` as a conversation AI assistant.

| Model | Schema Alignment | Clinical Reasoning | Overall |
|---|---|---|---|
| Llama-2-7b-chat-hf [Touvron et al., 2023] | $47.66 \pm 15.31$ | $47.55 \pm 11.73$ | $47.60 \pm 9.62$ |
| SynthIA-7B-v1.3 [Tissera, 2023] | $47.18 \pm 5.84$ | $49.16 \pm 4.99$ | $48.17 \pm 3.83$ |
| Mistral-7B-OpenOrca [Lian et al., 2023] | $51.75 \pm 8.20$ | $51.18 \pm 7.67$ | $51.46 \pm 5.60$ |
| Llama-3-8b-Instruct [Touvron et al., 2023] | $56.18 \pm 7.08$ | $55.07 \pm 7.25$ | $55.62 \pm 5.05$ |
| MPT-7b-8k-instruct [MosaicML, 2023] | $68.13 \pm 8.95$ | $53.90 \pm 4.92$ | $61.01 \pm 5.19$ |
| vicuna-7b-v1.5 [Chiang et al., 2023] | $66.81 \pm 5.61$ | $62.40 \pm 4.59$ | $64.60 \pm 3.63$ |
| dolphin-2.0-mistral-7b [Cognitive, 2023] | $63.06 \pm 5.36$ | $72.66 \pm 7.47$ | $67.86 \pm 4.64$ |
| `Llemr` + Stage 1 | $\mathbf{69.71 \pm 6.32}$ | $64.35 \pm 7.21$ | $67.03 \pm 6.83$ |
| `Llemr` + Stage 1&2 | $\mathbf{70.42 \pm 5.88}$ | $\mathbf{76.23 \pm 4.23}$ | $\mathbf{73.33 \pm 5.30}$ |

information extraction, enabling it to perform more advanced clinical reasoning tasks based on the given instructions.

# 6 Experiments

We conduct experiments to evaluate two key components: the quality of the generated clinical instruction-following data, and the performance of the proposed `Llemr`. We design experiments to answer the following research questions: (1) How well does `Llemr` perform as a conversational clinical AI assistant? (2) How does `Llemr` compare to SOTA methods on standard clinical predictive benchmarks?

## 6.1 Performance as Conversational AI Assistant

**Datasets.** To evaluate the performance of `Llemr` as a conversational clinical AI assistant, we construct a separate test set of 200 QA pairs. Specifically, we randomly select 100 ICU admissions from the previously held-out test set. These 100 admissions are unseen during training. Then, we follow the procedure described in Section 4 to generate 100 questions each for schema alignment and clinical reasoning.

**Metrics.** We follow Li et al. [2023a] and leverage GPT to quantify the quality of the generated responses. For each question, we generate a reference response with GPT-4. Then, we feed the question, the ground-truth answer, the GPT-4 generated answer, and the candidate LLM generated answer to GPT-3.5 and ask it to score both answers on a scale of 1 to 10, with higher scores indicating better performance. We prompt the GPT-3.5 to consider the helpfulness, relevance, accuracy, and level of detail of the responses. Next, we compute the relative score for the candidate LLM by normalizing against the GPT-4 reference score. We also report 95% confidence interval based on the 200 questions.

**Baselines.** We compare `Llemr` against seven popular open-sourced LLMs. For each baseline model, we concatenate the patient's event sequence and the corresponding question and input them directly into the LLM. These baseline LLMs have a context length from 4K to 32K tokens. For patients with long event sequences, we adopt Fleming et al. [2024]'s multi-step refinement approach. We segment the EHR sequence into chunks and process them sequentially. LLMs will generate an initial response from the first chunk, then decide whether to update or maintain this response as they process each subsequent chunk.

**Results.** The results are reported in Table 3. Overall, we can see that `Llemr` outperforms all existing LLM baselines and matches 73% of the GPT-4 performance. Surprisingly, we find that stage-1 training alone yields performance improvements, despite only the linear projection layer is trained. This may be due to `Llemr`'s architecture, which enables it to process the entire patient EHR sequence at once, whereas the baseline LLMs process it in chunks. Incorporating Stage 2 training further boosts performance, especially for clinical reasoning tasks.

In Table 4, we show examples of the generated responses and compare `Llemr` against its backbone LLM Vicuna [Chiang et al., 2023]. Generally, we find that `Llemr` tends to give more concise and

Table 4: Example of the generated response from Llemr.

| | Schema Alignment |
|---|---|
| User | When was the first Blood Anion Gap Chemistry measured? |
| Gold | 6.02 hours after admission. |
| Vicuna | The initial Blood Anion Gap Chemistry measurement was 14 mEq/L, which is within the normal range of 8-16 mEq/L. |
| Llemr | The measurement was taken at 6.02 hours. |
| User | Where was the patient admitted? |
| Gold | The patient was admitted to the emergency room. |
| Vicuna | The patient was admitted to the hospital with an emergency room complaint of agitation. The patient was transferred to the Medical Intensive Care Unit (MICU). The patient's diagnosis is not provided in the given data. |
| Llemr | The patient was initially admitted to the Emergency Room. |
| | Clinical Reasoning |
| User | What was the rationale for not pursuing surgery for the patient with severe c. diff and septic shock? |
| Gold | Surgery was not considered for the patient due to being a poor surgical candidate with severe lung disease and because surgery was not aligned with the patient's goals of care based on discussions with him and his family. |
| Vicuna | The patient was not pursued surgically due to concerns about the patient's overall condition and the potential risks of surgery. The patient's lab results showed abnormal values for various chemistry and hematology tests. |
| Llemr | The rationale for not pursuing surgery for the patient was likely due to the patient's critical condition and the presence of multiple lung diseases. |
| User | What antibiotic was used for perioperative coverage during the aortic valve replacement surgery? |
| Gold | Cefazolin was used for perioperative antibiotic coverage during the aortic valve replacement surgery. |
| Vicuna | The antibiotic used for perioperative coverage during the aortic valve replacement surgery was likely Omeprazole. |
| Llemr | The data only includes various medications such as analgesics, sedatives, and other treatments, but no antibiotics were prescribed specifically for the surgery. |

precise answers. In contrast, Vicuna [Chiang et al., 2023] tends to give broad responses or directly copy information from the input sequence. We also note that the last question contains information only from the discharge summaries. This can happen as the clinical reasoning data was generated from discharge summaries instead of EHR tables. This points out a potential future direction of further filtering the generated instruction-responses data.

## 6.2 Performance on Standard Clinical Predictive Benchmarks

**Datasets.** We leverage the held-out test set of 5585 ICU admissions from the MIMIC-IV database. The training and validation sets share the same cohorts as the ones used in instruction-tuning. Additional patient filtering is performed for each task introduced below.

**Tasks & Metrics.** We focus on four common clinical predictive tasks.

- **Mortality prediction** aims to predict whether the patient will pass away upon discharge using events from the first 48 hours of the hospital admission.

- **Length-of-stay prediction** aims to determine whether the patient's hospital stay will be longer than 7 days using the first 48 hours of the hospital admission. For both mortality and length-of-stay prediction tasks, patients with hospital length-of-stay less than 48 hours are filtered.

- **Readmission prediction** aims to predict whether the patient will be readmitted back to the hospital within 14 days following current discharge using all events from the current admission. Patients who are deceased in the current hospital admission are filtered for this task.

- **Diagnosis classification** aims to classify which acute care conditions are present using all events from the current admission. We follow existing works [Harutyunyan et al., 2019] and define disease labels with 25 conditions that are common in adult ICUs, including 12 critical conditions, (e.g.,

Table 5: Results on the MIMIC-IV clinical predictive benchmark tasks.

| Method | Mortality | Readmission | Length-of-Stay | Diagnosis |
|---|---|---|---|---|
| RNN [Cho et al., 2014] | 0.8002 (0.02) | 0.6643 (0.01) | 0.6833 (0.03) | 0.7735 (0.01) |
| Transformer [Vaswani et al., 2017] | 0.8241 (0.03) | 0.7006 (0.01) | 0.6990 (0.01) | 0.8025 (0.02) |
| RETAIN [Choi et al., 2016a] | 0.8302 (0.02) | 0.6994 (0.01) | 0.7015 (0.01) | 0.8073 (0.02) |
| GRASP [Zhang et al., 2021] | 0.8362 (0.01) | 0.7155 (0.01) | 0.7100 (0.03) | 0.8005 (0.02) |
| GenHPF [Hur et al., 2024] | 0.8258 (0.02) | 0.7102 (0.01) | 0.6993 (0.02) | 0.8103 (0.03) |
| REMed [Kim et al., 2024] | 0.8346 (0.02) | 0.7193 (0.02) | 0.7018 (0.01) | **0.8128 (0.01)** |
| Llemr (further-trained) | **0.8388 (0.01)** | **0.7251 (0.03)** | **0.7132 (0.01)** | 0.8086 (0.01) |

respiratory failure; 8 chronic conditions (e.g., diabetes); and 5 mixed (i.e., recurring or chronic with periodic acute episodes) conditions (e.g., cardiac dysrhythmias).

The first three tasks are binary classification and we calculate the Area Under the Receiver Operating Characteristic Curve (AUC-ROC) scores. The last task is multilabel classification and we report the macro AUC-ROC. We report the average scores and standard deviation by performing bootstrapping (i.e., sampling with replacement) 1000 times.

**Baselines.** We compare Llemr with three groups of baselines: (1) general machine learning (ML) methods: RNN [Cho et al., 2014] and Transformer [Vaswani et al., 2017]; (2) ML models specifically designed for clinical predictive modeling: RETAIN [Choi et al., 2016a], GRASP [Zhang et al., 2021], GenHPF [Hur et al., 2024]; (3) Foundation models that alleviate the need for manual preprocessing: REMed [Kim et al., 2024]. The first two baseline groups rely on hand-crafted features and thus we follow existing works [Zhang et al., 2021, Jiang et al., 2024] and only input the procedure, lab, and prescription features; and truncate event sequence to 512.

**Results.** It is important to note that these clinical predictive tasks are quite different from the instruction-following tasks. Thus, we perform an additional supervised fine-tuning step for Llemr. A new classification head (i.e., linear layer) is added on top of Llemr and trained for each clinical predictive task.

The results on clinical predictive benchmarks can be found in Table 5. Both Llemr and the baseline methods are supervised trained for the benchmark tasks. We select the best model weights on the validation set and report the performance on the test set. First, we can see that baselines relied on manual data engineering perform quite well, reaching 0.80 AUC-ROC score for mortality prediction and diagnosis classification. Among them, RETAIN [Choi et al., 2016a], GRASP [Zhang et al., 2021], and GenHPF [Hur et al., 2024] perform slightly better by injecting various domain-specific inductive biases. Interestingly, despite being free from feature engineering, REMed [Kim et al., 2024] performs even better than many other baselines. This is probably because REMed [Kim et al., 2024] is able to take a broader range of events as input and utilize them as additional information. Lastly, we can see that the fine-tuned Llemr can perform better or on par with SOTA methods. This demonstrates the flexibility and adaptability of Llemr.

## 7 Conclusion

The remarkable abilities of LLMs to understand complex inputs and follow instructions for diverse tasks suggest their potential to simplify and enhance the analysis of EHRs. However, developing a conversational AI assistant for EHR data is difficult due to the following challenges: (1) the lack of large-scale instruction tuning data and (2) the limitation of model architectures in handling EHR data with complex schemas. In this paper, we introduce MIMIC-Instr, a dataset of over 400K open-ended instruction-tuning examples generated by GPT-3.5. This dataset covers a broad range of topics and can be used to instruction-tune general-purpose LLMs. Additionally, we propose Llemr, a generic framework designed to empower LLMs to encode EHR data with heterogeneous schema. Evaluation results show that Llemr exhibits excellent capabilities in answering diverse inquiries about a patient and performs on par with SOTA baselines when further fine-tuned for clinical predictive tasks.

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
