# A Datasheet for `MIMIC-Instr`

## A.1 Motivation

- **For what purpose was the dataset created?**

  We created `MIMIC-Instr` to enable instruction tuning LLMs to understand EHR data. The end goal is to develop conversational AI assistants to help physicians in information extraction and clinical reasoning on the EHR data.

- **Who created the dataset (e.g., which team, research group) and on behalf of which entity (e.g., company, institution, organization)?**

  The `MIMIC-Instr` dataset was created by the authors of this paper.

- **Who funded the creation of the dataset?**

  To be added upon publication.

## A.2 Composition

- **What do the instances that comprise the dataset represent (e.g., documents, photos, people, countries)?**

  `MIMIC-Instr` contains natural language instructions (questions) and responses (answers). Each QA pair is matched to a patient record from the publicly available MIMIC-IV [Johnson et al., 2023] database.

- **How many instances are there in total (of each type, if appropriate)?**

  There are 353,448 QA pairs in the schema alignment subset and 50,188 QA pairs in the clinical reasoning subset. Together, there are 403,636 QA pairs.

- **Does the dataset contain all possible instances or is it a sample (not necessarily random) of instances from a larger set?**

  All instances were provided.

- **What data does each instance consist of?**

  Each instance consists of a subject id (integer), a hospital admission id (integer), a question (str), and an answer (str). The subject id and hospital admission id can be used to locate the corresponding patient record in the MIMIC-IV [Johnson et al., 2023] database.

- **Is there a label or target associated with each instance?**

  The answer acts as the label.

- **Is any information missing from individual instances?**

  No.

- **Is any information missing from individual instances?**

  No.

- **Are there recommended data splits (e.g., training, development/validation, testing)?**

  Yes. The data splits will be provided.

- **Are there any errors, sources of noise, or redundancies in the dataset?**

  The schema alignment subset was generated based on hand-crafted templates and then paraphrased by GPT-3.5. The clinical reasoning subset was directly generated by GPT-3.5. The questions and results can contain errors, noise, and redundancies.

- **Is the dataset self-contained, or does it link to or otherwise rely on external resources (e.g., websites, tweets, other datasets)?**

  The dataset needs to be used with the publicly available MIMIC-IV [Johnson et al., 2023] database.

- **Does the dataset contain data that might be considered confidential (e.g., data that is protected by legal privilege or by doctor–patient confidentiality, data that includes the content of individuals' non-public communications)?**

  No. The dataset was constructed based on the MIMIC-IV [Johnson et al., 2023] database which is de-identified. We used Azure's HIPAA-compliant OpenAI API in accordance with PhysioNet's regulations.

- **Does the dataset contain data that, if viewed directly, might be offensive, insulting, threatening, or might otherwise cause anxiety?**

  No.

- **Does the dataset identify any subpopulations (e.g., by age, gender)?**

  Yes. Some generated questions may ask about patient's age, gender, and ethnicity. These demographic information were provided in the MIMIC-IV database and went through the de-identification process to avoid identifying individual patients.

- **Is it possible to identify individuals (i.e., one or more natural persons), either directly or indirectly (i.e., in combination with other data) from the dataset?**

  No.

- **Does the dataset contain data that might be considered sensitive in any way (e.g., data that reveals race or ethnic origins, sexual orientations, religious beliefs, political opinions or union memberships, or locations; financial or health data; biometric or genetic data; forms of government identification, such as social security numbers; criminal history)?**

  No. The source datasets are already de-identified.

### A.3 Collection process

- **How was the data associated with each instance acquired?**

  The schema alignment subset was generated based on template and then paraphrased by GPT-3.5. The clinical reasoning subset was directly generated by GPT-3.5.

- **What mechanisms or procedures were used to collect the data (e.g., hardware apparatuses or sensors, manual human curation, software programs, software APIs)?**

  The question templates were designed manually using EHRSQL [Lee et al., 2023] as a reference. The GPT-3.5 API was from Azure's HIPAA-compliant platform.

- **If the dataset is a sample from a larger set, what was the sampling strategy (e.g., deterministic, probabilistic with specific sampling probabilities)?**

  Not applicable.

- **Who was involved in the data collection process (e.g., students, crowd workers, contractors) and how were they compensated (e.g., how much were crowd workers paid)?**

  The `MIMIC-Instr` dataset was created by the authors of this paper. No crowd workers were involved.

- **Over what timeframe was the data collected?**

  The MIMIC-IV [Johnson et al., 2023] data was collected between 2008 and 2019. The `MIMIC-Instr` dataset was built upon MIMIC-IV in 2024.

- **Were any ethical review processes conducted (e.g., by an institutional review board)?**

  Not applicable.

- **Did you collect the data from the individuals in question directly, or obtain it via third parties or other sources (e.g., websites)?**

  The `MIMIC-Instr` dataset was built upon the MIMIC-IV [Johnson et al., 2023] database.

- **Were the individuals in question notified about the data collection?**

  Not applicable.

- **Did the individuals in question consent to the collection and use of their data?**

  Not applicable.

- **If consent was obtained, were the consenting individuals provided with a mechanism to revoke their consent in the future or for certain uses?**

  Not applicable.

- **Has an analysis of the potential impact of the dataset and its use on data subjects (e.g., a data protection impact analysis) been conducted?**

  Not applicable.

### A.4 Preprocessing/cleaning/labeling

- **Was any preprocessing/cleaning/labeling of the data done (e.g., discretization or bucketing, tokenization, part-of-speech tagging, SIFT feature extraction, removal of instances, processing of missing values)?**
  Not applicable.

- **Was the "raw" data saved in addition to the preprocessed/cleaned/labeled data (e.g., to support unanticipated future uses)?**
  Not applicable.

- **Is the software that was used to preprocess/clean/label the data available?**
  Yes. We utilized Python, Jupyter Notebooks, and OpenAI API.

### A.5 Uses

- **Has the dataset been used for any tasks already?**
  Currently, the dataset is used for instruction tuning LLMs to understand EHR data.

- **Is there a repository that links to any or all papers or systems that use the dataset?**
  No.

- **What (other) tasks could the dataset be used for?**
  Question answering based on EHR data.

- **Is there anything about the composition of the dataset or the way it was collected and preprocessed/cleaned/labeled that might impact future uses?**
  This dataset was mainly generated with templates or by GPT-3.5. So it could contain errors and noise.

- **Are there tasks for which the dataset should not be used?**
  Not applicable.

### A.6 Distribution

- **Will the dataset be distributed to third parties outside of the entity (e.g., company, institution, organization) on behalf of which the dataset was created?**
  Yes.

- **How will the dataset will be distributed (e.g., tarball on website, API, GitHub)?**
  The dataset will be released via PhysioNet upon publication.

- **When will the dataset be distributed?**
  Upon publication.

- **Will the dataset be distributed under a copyright or other intellectual property (IP) license, and/or under applicable terms of use (ToU)?**
  The dataset will be distributed under the PhysioNet Credentialed Health Data License: `https://www.physionet.org/content/ehr-ds-qa/view-license/1.0.0/`.

- **Have any third parties imposed IP-based or other restrictions on the data associated with the instances?**
  `MIMIC-Instr` needs to be used with the MIMIC-IV [Johnson et al., 2023] database which is under the PhysioNet Credentialed Health Data License: `https://www.physionet.org/content/ehr-ds-qa/view-license/1.0.0/`.

- **Do any export controls or other regulatory restrictions apply to the dataset or to individual instances?**
  No.

### A.7 Maintenance

- **Who will be supporting/hosting/maintaining the dataset?**
  The authors of this paper.

- **How can the owner/curator/manager of the dataset be contacted (e.g., email address)?**
  Contact the first/corresponding authors via email (to be added upon publication) or raise GitHub issue.

- **Is there an erratum?**
  No.

- **Will the dataset be updated (e.g., to correct labeling erros, add new instances, delete instances)?**
  Yes, we plan to update the datasets as we further improve the templates and OpenAI updates its API.

- **If the dataset relates to people, are there applicable limits on the retention of the data associated with the instances (e.g., were the individuals in question told that their data would be retained for a fixed period of time and then deleted)?**
  No.

- **Will older versions of the dataset continue to be supported/hosted/maintained?**
  Yes. This dataset has very simple schema, so the older versions can be easily supported.

- **If others want to extend/augment/build on/contribute to the dataset, is there a mechanism for them to do so?**
  Contact the first/corresponding authors via email (to be added upon publication) or raise GitHub issue.

## B   Use of OpenAI API

We used Azure's HIPAA-compliant platform in accordance with PhysioNet's regulations. We used the "gpt-35-turbo (0125)" version of GPT-3.5 and the "gpt-4 (0125-Preview)" version of GPT-4.

## C   Additional Information on the `MIMIC-Instr`

### C.1   Questions Templates

We provide all the question templates used in generating the schema alignment subset in Table 6

Table 6: Questions templates used for generating the schema alignment subset of `MIMIC-Instr`. The definition of the {time_period} keyword can be found in Table 7.

| Table | Template |
|---|---|
| patients & admissions | What was the gender of the patient? 
 What was the age of the patient? 
 What was the race of the patient? 
 What was the insurance of the patient? 
 What was the marital status of the patient? 
 What was the admission type of the patient? 
 What was the admission location of the patient? 
 What aws the chief complaint of the patient? |
| diagnoses_icd | What were the billled diagnoses of the patient? 
 What were the top five billled diagnoses of the patient? |
| labevents | What was the {measurement_name} at the {timestamp} hour? 
 Was the {measurement_name} at the {timestamp} hour normal? 
 What {category} measurements were performed on the {fluid} specimen at the {timestamp} hour? 
 What {category} measurements on the {fluid} specimen were abnormal at the {timestamp} hour? 
 What was the {first/last} {measurement_name} {time_period}? |

| | |
|---|---|
| | When was the {first/last} {measurement_name} {time_period}?
How many times did the patient have the {measurement_name} {time_period}?
What was the {maximum, minimum, average} {measurement_name} {time_period}?
Did the patient have any {measurement_name} {time_period}? |
| microbiologyevents | What microbiology tests were performed on the {spec_type_desc} specimen at the {timestamp} hour?
What organisms were found on the {spec_type_desc} specimen at the {timestamp} hour?
What were the antibiotics test results against the {org_name} on the {spec_type_desc} specimen at the {timestamp} hour?
Did the patient have any microbiology test on the {spec_type_desc} specimen {time_period}? |
| prescription | What was the composition of the prescribed {drug} at the {timestamp} hour?
What was the dose of the prescribed {drug} at the {timestamp} hour?
What was the administration route of the prescribed {drug} at the {timestamp} hour?
What was the administration duration of the prescribed {drug} at the {timestamp} hour?
What drugs were prescribed at the {timestamp} hour?
What was the composition of the {first/last} prescribed {drug} {time_period}?
What was the dose of the {first/last} prescribed {drug} {time_period}?
What was the administration route of the {first/last} prescribed {drug} {time_period}?
What was the administration duration of the {first/last} prescribed {drug} {time_period}?
When was the {first/last} {drug} prescription {time_period}?
How many times did the patient have the {drug} prescription {time_period}?
Was the patient prescribed with any {drug}? |
| transfers | Which unit was the patient transferred to at the {timestamp} hour?
When was the patient discharged from the hospital?
How long was the length of hospital stay of the patient in hours?
How long was the length of hospital stay of the patient in days? |
| inputevents | What was the amount of the IV administration {label} at the {timestamp} hour?
What was the duration of IV administration {label} at the {timestamp} hour?
What drugs were administered through IV at the {timestamp} hour?
What was the amount of the {first/last} IV administration {label} {time_period}?
What was the duration of the {first/last} IV administration {label} {time_period}?
When was the {first/last} {label} IV administration {time_period}?
How many times did the patient have the {label} IV administration {time_period}?
Was the patient administered with any {label} through IV {time_period}? |
| outputevents | What was the amount of the output {label} at the {timestamp} hour?
What was the amount of the {first/last} output {label} {time_period}?
What was the total amount of the output {label} {time_period}?
What was the {maximum, minimum, average} amount of the output {label} {time_period}?
When was the {first/last} output {label} {time_period}? |

| | How many times did the patient have the {label} output {time_period}? |
|---|---|
| | Did the patient have any {label} output {time_period}? |
| procedureevents | What procedures were performed at the {timestamp} hour? |
| | What procedures were performed {time_period}? |
| | What was the duration of the {first/last} {label} procedure {time_period}? |
| | When was the {first/last} {label} procedure {time_period}? |
| | How many times did the patient undergo the {label} procedure {time_period}? |
| | Did the patient undergo any {label} procedure {time_period}? |

Table 7: Definition of the {time_period} keyword.

| Keyword | Realization |
|---|---|
| {time_period} | during the first 12 hours |
| | during the first 24 hours |
| | during the first 48 hours |
| | during the last 12 hours |
| | during the last 24 hours |
| | during the last 48 hours |
| | during day day |
| | during the entire stay |

## C.2  Prompts

The prompts used for paraphrasing and generating the QA pairs can be found in Prompts 1 and 2.

---

**Prompt 1** Prompting GPT-3.5 to paraphrase the QA pairs generated with templates.

You are an AI assistant with expertise in medical knowledge.

Your input consists of a question-answer pair created using predefined rules.

Your primary task is to rephrase both the question and the answer to introduce variety in the wording while preserving their original meanings.

Objective:
1. Paraphrase both the question and the answer.
2. Ensure the paraphrased text is grammatically correct.
3. Adjust capitalization as needed
4. Maintain the original intent and meaning of the question-answer pair.
5. Format your response as follows:
- Question: [Your paraphrased question]
- Answer: [Your paraphrased answer]
6. Aim for brevity in both the question and answer.

Question: {input question}
Answer: {input question}

---

## C.3  MIMIC-IV Preparation

We construct our cohort using the ICU patients from the MIMIC-IV [Johnson et al., 2023] database. This database was constructed from the patients admitted to the ICU in the Beth Israel Deaconess Medical Center. This database covers 50920 patients with 66239 hospital admissions and 73181 ICU stays. We filter out patients without discharge summaries (909 admissions

removed), with more than two ICU stays per hospital admission (5,762 admissions removed), less than 18 years old (no admission removed), and with negative ICU/hospital length-of-stay (55 admissions removed). We then select the following tables from MIMIC-IV: hosp/patients, hosp/admissions, hosp/diagnosis, hosp/labevents, hosp/microbiologyevents, hosp/prescriptions, hosp/transfers, icu/icustays, icu/inputevents, icu/outputevents, icu/procedureevents. Note that icu/chartevents table is excluded due to two reasons: it contains bedside monitor data, which is very dense and should be better treated as timeseries instead of event sequence, and it has large overlapping information with other tables like hosp/labevents [Johnson et al., 2023]. Lastly, we remove patients without any generated QA, i.e. rejected by GPT-3.5 (2,987 admissions removed) and with more than 4,000 events (680 admissions removed). In the end, we have a total of 55846 admissions. We hold out 10% each for validation (5579) and testing (5585). The average number of events per patient admission is $559 \pm 543$.

We covert each event into text using the templates in Table 8.

Table 8: Templates used in converting event to text.

| Table | Template |
|---|---|
| patients & admissions | gender: {gender}, age: {age}, race: {race}, marital status: {marital_status}, insurance: {insurance}, type: {admission_type}, location: {admission_location}, chief complaint: {chief_complaint} |
| diagnoses_icd | Billed diagnosis: {long_title} |
| labevents | {fluid} {label} {category}: {value} {normal/abnormal} |
| microbiologyevents | {test_name} on {spec_type_desc}, organism grew: {org_name}, antibiotic tested: {ab_name}, antibiotic sensitivity: {interpretation}, comments: {comments} |
| prescriptions | {drug} ({prod_strength}), prescribed dose: {dose_val_rx} {dose_unit_rx}, route: {route}, duration: {duration} hour |
| transfers | {eventtype} to {careunit} |
| inputevents | {label} {amount} {amountuom} for {duration} hour |
| outputevents | {label} {value} {valueuom} |
| procedureevents | {label} for {duration} hour |

## C.4 Examples

Table 9: Randomly selected examples from the `MIMIC-Instr` dataset.

| Question | Answer |
|---|---|
| **Schema Alignment** | |
| What was the blood gas measurement obtained through intubation at 1.55 hours? | Intubated. |
| How was the Calcium Gluconate administered at 1.20 hour? | Intravenously (IV) |
| Was there any value in using an Assistance Device for the patient on day 4? | None. |
| What type of admission did the patient have? | Urgent |
| Did the patient receive any IV administration of NaCl 0.9% within the initial 48 hours? | Affirmative. |
| What type of insurance did the patient have? | The patient had Medicare coverage. |
| Did the patient receive any IV administration of NaCl 0.9% on day 13? | Negative. |
| What was the timing of the initial blood PT Hematology measurement within the past 48 hours? | The measurement was taken at 38.85 hours. |
| What was the highest volume of fluid drained by Chest Tube #1 throughout the patient's stay? | The maximum amount of fluid drained by Chest Tube #1 was 40.00 ml. |
| How much fluid did Chest Tube #1 produce in the initial 12-hour period? | The output from Chest Tube #1 was 80.00 ml. |
| How much Foley output was typically observed on the second day? | The average amount of Foley output on day 2 was 22.50 ml. |
| How old was the individual? | 63 years old. |
| How much NaCl 0.9% was administered intravenously at 63.17 hours? | The IV administration of NaCl 0.9% was 100.00 ml. |
| How much was the lowest output from the Foley catheter in the past 48 hours? | The minimum output from the Foley catheter was 20.00 ml. |
| What was the primary reason for the patient's visit? | The patient experienced discomfort in the chest. |
| What was the blood bicarbonate chemistry measurement within the initial 24-hour period? | The blood bicarbonate chemistry measurement was 23.0 mEq/L. |
| What was the timing of the initial CT scan within the first 12 hours? | The CT scan was performed at 2.68 hours. |
| Where was the patient admitted? | The patient was admitted to the emergency room. |
| What was the sex of the individual? | Female |
| What was the timing of the most recent Ultrasound procedure on day 1? | It occurred at 4.68 hours. |
| **Clinical Reasoning** | |
| What was the reason for discontinuing anticoagulation therapy in the patient with pericardial tamponade post-AVR surgery? | Anticoagulation therapy was discontinued in the patient with pericardial tamponade post-AVR surgery due to a super-therapeutic INR of 6.8 and no clear indication for anticoagulation at that time. |
| What was the intervention performed during the endoscopy for the patient with a foreign body in the esophagus? | The foreign body of food was removed during the endoscopy, and no other intervention was done. |
| What medications were initiated for the patient in the ICU due to frequent irregular heart rate and atrial ectopies? | The patient was started on beta blocker and Amiodarone for frequent irregular heart rate and atrial ectopies while in the ICU. |
| What interventions were performed on the patient during the procedure in the Operating Room? | The patient underwent CABG x 4 (coronary artery bypass grafting) and AVR (aortic valve replacement) with Dr. [Doctor's Name]. |

| | |
|---|---|
| What interventions were performed to manage the patient's ventricular tachycardia? | The patient's ventricular tachycardia was managed with Amiodarone boluses, electrolyte repletion, and defibrillation. EP recommended continuing Amiodarone for treatment. |
| What was the patient's hematocrit level upon admission to the Acute Care Trauma Surgery service in the intensive care unit? | The patient's hematocrit was 39.6 upon admission to the Acute Care Trauma Surgery service in the intensive care unit. |
| What was the reason for not starting the patient on coumadin despite having atrial fibrillation postoperatively? | The patient was not started on coumadin due to the limited time frame of atrial fibrillation. |
| What prompted the initiation of Zosyn and a bronchoscopy for the patient during his hospital stay? | The patient had multiple episodes of PO intolerance complicated by aspiration and desaturation, leading to the initiation of Zosyn and a bronchoscopy to further evaluate his airway. |
| What was the patient's initial presentation upon transfer from rehab that raised concern for aortic valve thrombosis? | The patient presented with dyspnea and weight gain, initially requiring BiPAP, which raised concern for aortic valve thrombosis. |
| What was the reason for the patient's emergent surgery in the operating room? | The patient underwent Emergent repair of type A aortic dissection with cardiac tamponade, ascending aorta and Hemiarch replacement with 28mm Gelweave graft under circulatory arrest. |
| What was the reason for the patient's admission to the ICU and what was the finding on EGD? | The patient was admitted to the ICU due to difficulty managing secretions, and EGD revealed a piece of chicken impacted in the mid esophagus that was advanced into the stomach. |
| What was the reason for holding the patient's Plavix for 7 days according to Neurosurgery? | Neurosurgery recommended holding her Plavix for 7 days due to stable SAH and no need for seizure prophylaxis. |
| What was the reason for the patient's transfer to the ICU from the rehabilitation facility? | The patient was transferred to the ICU from the rehabilitation facility for Tylenol overdose in fulminant liver failure most likely due to Tylenol overdose with a Tylenol level of 154 at 24 hours from ingestion event. |
| What was the reason for consulting Ophthalmology for the patient on post operative day 1? | Ophthalmology was consulted for the patient's complaints of bilateral floaters to assess for signs of hemorrhage or neovascularization. |
| What interventions were implemented to manage the patient's post-operative Atrial Fibrillation? | The patient was started on Amiodarone and Coumadin after experiencing several hours of postoperative Atrial Fibrillation on POD 4. |
| What specialties evaluated the patient upon arrival and found no acute interventions necessary? | Plastics and ophthalmology evaluated the patient upon arrival and found no acute interventions necessary. |
| What was the reason for the patient's elevated white blood cell count post-operatively? | The elevated white blood cell count post-operatively was thought to be due to the patient's decadron. |
| What was the reason for holding certain home medications, including methadone, for the patient during their ICU course? | The patient's home medications that can lead to sedation, including methadone, were held due to findings of generalized spike wave and left hemisphere discharges on EEG, as well as an event with right gaze deviation and body stiffening, unresponsiveness. |
| What was the patient's LVEF and LV systolic function according to the TTE results? | LVEF 47%, moderate regional LV systolic dysfunction in setting of overall mild global LV systolic dysfunction c/w prior myocardial infarction in the mid LAD territory. |

| | |
|---|---|
| What was the patient's neurological status on the day of discharge to rehabilitation? | On [date of discharge], the patient was neurologically intact, afebrile, ambulating with assistance, tolerating a diet, voiding and stooling without difficulty, and his pain was well controlled with oral pain medications. |

# D   Additional Experimental Setup for Conversational AI Assistant

Prompts for instructing GPT-3.5 to evaluate the generated response can be found in Prompt 3. This prompt was adapted from the one used in LLaVA-Med [Li et al., 2023a].

---

**Prompt 3** Prompting GPT-3.5 to evaluate the generated response.

You are a helpful and precise assistant for evaluating the quality of responses.

Please assess the performance of two clinical AI assistants based on the question and the ground-truth answer provided below.

Your evaluation should consider helpfulness, relevance, accuracy, and level of detail.

Rate each AI assistant's response with a single score on a scale of 1 to 10, where 10 represents excellent performance.

Please first output a single line containing only two values indicating the scores for Assistant 1 and 2, respectively. The two scores are separated by a space.

In the subsequent line, provide a concise explanation of your evaluation.

Avoid any potential bias and ensure that the order in which the responses were presented does not affect your judgment.

Question
{input question}
End of Question

Ground-truth Answer
{input ground-truth answer}
End of Ground-truth Answer

Assistant 1 Answer
{input assistant 1 answer}
End of Assistant 1 Answer

Assistant 2 Answer
{input assistant 2 answer}
End of Assistant 2 Answer

---

# E   Additional Experimental Setup for Clinical Predictive Benchmarks

We leverage the held-out test set of 5585 ICU admissions from the MIMIC-IV database. The training and validation set share the same cohorts as the ones used in instruction-tuning. Additional patient filtering is performed for each task introduced below.

**Mortality prediction** aims to predict whether the patient will pass away upon discharge using events from the first 48 hours of the hospital admission. Patients with hospital length-of-stay less than 48 hours are filtered. The prevalence rate is 0.10.

**Length-of-stay prediction** aims to determine whether the patient's hospital stay will be longer than 7 days using the first 48 hours of the hospital admission. Patients with hospital length-of-stay less than 48 hours are filtered. The prevalence rate is 0.18.

**Readmission prediction** aims to predict whether the patient will be readmitted back to the hospital within 14 days following current discharge using all events from the current admission. Patients who are deceased in the current hospital admission are filtered for this task. The prevalence rate is 0.11.

**Diagnosis classification** aims to classify which acute care conditions are present using all events from the current admission. We follow existing works [Harutyunyan et al., 2019] and define disease labels with 25 conditions that are common in adult ICUs, including 12 critical conditions, (e.g., respiratory failure; 8 chronic conditions (e.g., diabetes); and 5 mixed (i.e., recurring or chronic with periodic acute episodes) conditions (e.g., cardiac dysrhythmias). The complete list of disease labels and prevalence rates can be found in Table 10.

Table 10: List of all 25 disease labels.

| Disease | Category | Prevalence |
|---|---|---|
| Septicemia | acute | 0.15 |
| Fluid and electrolyte disorders | acute | 0.38 |
| Acute myocardial infarction | acute | 0.08 |
| Congestive heart failure | acute | 0.25 |
| Acute cerebrovascular disease | acute | 0.09 |
| Pneumonia | acute | 0.13 |
| Pleurisy; pneumothorax; pulmonary collapse | acute | 0.09 |
| Respiratory failure | acute | 0.24 |
| Other lower respiratory disease | acute | 0.12 |
| Other upper respiratory disease | acute | 0.05 |
| Other liver diseases | acute | 0.13 |
| Gastrointestinal hemorrhage | acute | 0.06 |
| Acute and unspecified renal failure | acute | 0.27 |
| Complications of surgical procedures | acute | 0.21 |
| Shock | acute | 0.12 |
| Diabetes mellitus without complication | chronic | 0.18 |
| Diabetes mellitus with complications | chronic | 0.14 |
| Disorders of lipid metabolism | chronic | 0.41 |
| Essential hypertension | chronic | 0.41 |
| Hypertension with complications | chronic | 0.21 |
| Coronary atherosclerosis and other heart disease | chronic | 0.32 |
| Conduction disorders | chronic | 0.10 |
| Cardiac dysrhythmias | chronic | 0.36 |
| Chronic obstructive pulmonary disease | chronic | 0.14 |
| Chronic kidney disease | chronic | 0.20 |

# F   Limitations and Broader Impacts

The `MIMIC-Instr` dataset was generated based on hand-crafted templates and with the help of GPT-3.5. This means the dataset could (and probably would) contain error and noise. The inherent inaccuracies stemming from the data generation process could introduce biases or distort the models' understanding of real-world clinical scenarios. Consequently, when applying these models in the real-world setting, one must carefully evaluate their performance and interpretability to ensure that they reliably support clinical decision-making. Furthermore, future work may focus on refining data generation methods to improve the quality and realism of training datasets, thus enhancing the models' applicability and accuracy in practical healthcare applications.