# OpenReview forum: "Instruction Tuning Large Language Models to Understand Electronic Health Records"
_NeurIPS.cc/2024/Datasets_and_Benchmarks_Track — NeurIPS 2024 Track Datasets and Benchmarks Spotlight_

### Official Review · Reviewer_Tq86 · 2024-06-17

**Rating:** 9
**Confidence:** 3
**Correctness:** I think it is correct.
**Clarity:** I think it is well-written in general.

**Review:**

The paper has a significant impact on the community in that it is the one of the very first high-quality clinical dataset created for LLM training and evaluation. This dataset is sourced from MIMIC, one of the most well-known open-source dataset in the world. It will benefit and empower our community, creating new opportunities to embrace the LLM era even for clinical tasks, most of which are not suitable for LLM. It will create a novel paradigm to deal with clinical prediction tasks in the future, which distinguishes itself from most of the previous works.

**Strengths:**

1. Great dataset for general clinical understanding and prediction.
2. Broad community impact.
3. Outstanding/promising results.

**Additional Feedback:**

It would be great for the authors to spend some time in patience to make this work more reader-friendly to the audience, e.g., spend some time to make the info in the link easier to understand, provide more details, etc. I think it would be good impact.

**Documentation:**

Yes, links are provided.

**Ethics:**

No concern.

**Limitations:**

I do not see discussion on limitations of this work. I highly and definitely recommend the authors to comment on limitations and provide insights on future discoveries, since this paper is definitely a good paper for publication already.

**Opportunities For Improvement:**

The main opportunities for improvement with this manuscript are two-fold. First, the description about the dataset, such as the statistics, is now sufficient enough. We do not know how the 350k schema alignment data is generated and sampled: how many of them are about asking the demographics, how many of them are about asking the lab results, how many patients are with heart/kidney/cancer/diabetes problems? Second, we only know that Llemr is a language model, with text output. How do you know whether it is correct about mortality prediction, diagnosis, etc.? Is there a program to detect whether the text response is correct and how? What about treatment recommendation task? There is currently no code for your Llemr, so I cannot check how you did that and reproduce the result. Please do release the code since it would be helpful for me to assess the quality/reliability of the result -- Table 5 looks too good to be true, and I genuinely want to believe this is true. It would be a huge impact in my eyes, but the audience would like to know more details. I will re-evaluate after the rebuttal.

There are also some minor points. Why not use ChatGPT-4o and GPT-4 as baselines with proper prompt engineering? It is totally possible even if we cannot train or finetune a GPT-4. Also, how do you know the answer to the clinical reasoning, since some answers seem unlikely to be in the EHR data itself? For example, the actual ground-truth answer to "What is the recommended dose for levalbuterol for the patient’s severe COPD exacerbation?" could actually be a range instead of any certain values recorded in the EHR. It is hard to say that the recommended dose is absolutely 2, and 3 is incorrect. This is about realistic reasoning, and the best treatment may not necessarily be based on what happened to that patient in the real world. How do you determine that if we do not have it in the EHR data? I think there needs more explanation if I do not understand something wrongly.

**Relation To Prior Work:**

Yes, most important prior works are mentioned.

**Summary And Contributions:**

The paper introduced a 400k large-scale dataset sourced from MIMIC-III for empowering LLMs to perform information extraction and clinical reasoning on EHR data. A proposed Llemr foundation model, pretrained on Schema Alignment and Clinical Reasoning tasks, was also introduced, which outperforms various baselines of LLMs and clinical prediction models.

---

> ### Author Rebuttal · Authors · 2024-08-24
>
> We appreciate the detailed feedback and the positive evaluation of our work. We are particularly grateful for the acknowledgment of the significance and potential impact of our proposed dataset and the Llemr foundation model. Below, we address the specific concerns and suggestions raised.
>
> ## 1. How is the alignment data generated and sampled?
>
> For each patient, we randomly and uniformly sampled 6 tables out of `hosp/patients`, `hosp/admissions`,  `hosp/diagnosis`, `hosp/labevents`, `hosp/microbiologyevents`, `hosp/prescriptions`, `hosp/transfers`,  `icu/icustays`, `icu/inputevents`, `icu/outputevents`, `icu/procedureevents`. For each sampled table, we randomly and uniformly selected one template.
>
> ## 2. How does Llemr work for clinical predictive tasks?
>
> We would like to clarify that we perform an additional supervised fine-tuning step for each of the clinical predictive tasks using the `AutoModelForSequenceClassification` API from Hugging Face. A new classification head (i.e., linear layer) is added on top of Llemr and trained for the clinical predictive tasks.
>
> ## 3. Code and Dataset Release
>
> For the code, we are in the final stages of preparing our codebase for release. For the data, due to the user agreements of the MIIMC-IV data, we need to work with PhysioNet in releasing the MIMIC-Instr dataset. We are committed to making both code and data available on PhysioNet upon acceptance.
>
> ## 4. Using ChatGPT-4o or GPT-4 as baselines
>
> We would like to clarify that we have used GPT-4 as a baseline during the evaluation. Specifically, to assess the quality of the responses, we ask GPT-3.5 to score both the Llemr-generated answer and the GPT-4 generated answer against the ground-truth answer and report the relative performance of Llemr by calculating the final score as the ratio of the Llemr score to the GPT-4 score.  This allows for a more consistent and reliable comparison between models.
>
> ## 5. Generated Questions Might be Unanswerable
>
> We acknowledge that the questions generated from discharge summaries may sometimes require context that cannot be fully captured by the structured EHR data alone. This is in fact a common challenge when working with synthetic data. However, our primary objective with this work is to provide a proof of concept, demonstrating the feasibility of using large language models to reason over structured EHR data.
>
> ## 6. Discussion on Limitations
> We include the limitation section in the appendix. We will make sure to add a reference to the limitation section in the main paper.

---

### Official Review · Reviewer_vJma · 2024-07-19
**Review of instruction tuning for EHRs**

**Rating:** 9
**Confidence:** 5
**Clarity:** Yes

**Review:**

The authors adopt the Medical Event Data Standard, which forms a tuple of event time, event name, and event value. These tuples can be represented as natural senteces. In theory these representations should be portable across EHR systems. The authors use ClinicalBERT to embed these sentences.

The first phase of training involved using QA about the EHR sequences to train a layer to project frozen clinicalBERT embeddings into the natural language space.
Q1) why was clinicalBERT frozen? The number of paramaters seem miniscule compared to fine-tuning the remaining LLM.

Upon concatenating the adapted events as tokens in the LLM's representation, a decoder only fine-tuned version of vicuna. Schema alignment alone (stage 1) showed modest improvement over vicuna, since it was able to ingest the entire EHR sequence instead of chunks. However fine-tuning on the stage 2 contextual questions improved the schema alignment and clinical reasoning scores. Of note, these compare to a GPT-4 benchmark, not a human benchmark

For the performance on standard clinical benchmarks, only the first 48 hours of data are used. These data go on to predict in-hospital mortality, length of stay prediction, readmission within 14 days, and ICD10 diagnosis. Crucially, the authors fail to implement a gap time between the end of data collection and the prediction. Fortunately this is a fast fix. Please eliminate test subjects where they experience an event within 50 hours (a gap time of 2 hours between the end of the data and the start of prediction). This also ensures that the charttime of the labs and vitals measured at hour 48 would feasibly be in the EHR by the time the prediction is conducted as well. For non-ICU patients a longer gap time is recommended.

One of the disadvantages in the baseline models are that they fail to ingest long sequences of data. It would be desirable to have an imminent mortality task or imminent decompensation task as conducted in Harutyunyan et al 2019. How does this compare to the commonly used clinical algorithm NEWS2?

One of the claims with Llmer is that it is EHR agnostic, and can span different EHRS. While clinicalBERT is trained on MIMIC-III and not fine-tuned in this paper, the adapting layer is fine-tuned, which may be overfit to the domain attributes of MIMIC-IV. It would be desirable to reproduce table 4 on held out sites, rather than on held out patients at the same site. This would allow for better comparisons between GenHPF, and FHIR embeddings. eICU would be a good candidate, as it is still in the physionet ecosystem and was not generated at the same hospital.

The medical event data standard involves encoding timestamps using clinicalBERT. How do constant shifts in the timestamp affect performance? Ideally the model should be insensitive to shifts.

**Strengths:**

The training curriculum seems to generate pragmatic results.
Zero shot performance is competetive on in-domain clinical tasks

**Additional Feedback:**

n/a

**Correctness:**

It sounds correct. The note about using Azure's HIPPA compliant platform for physionet data should be more prominent. If others wish to reproduce or extrapolate the methods to new datasets they should be warned that there are restrictions on the available platforms.

**Documentation:**

Yes.  MIMIC-Instr will be made available on physionet.

**Ethics:**

They used a HIPPA compliant server. The note about using Azure's HIPPA compliant platform for physionet data should be more prominent. If others wish to reproduce or extrapolate the methods to new datasets they should be warned that there are restrictions on the available platforms.

**Limitations:**

The limitations of the paper largely fall in line with the opportunities for improvement, many of which can be easily addressed.
The largest limitation is that this study is done at a single site. Any potential benefits of having a Medical Event Standard format for generalising across EHRs has gone untested.

**Opportunities For Improvement:**

The authors had not included a gap time in their clinical task benchmark which may lead to label leakage.
ClinicalBert was not fine-tuned for this task despite not being accustomed to the Medical Event Standard format.
Include an imminent mortality task. LLM-based sequence models have yet to be evaluated for tasks requiring long contexts. It also helps benchmark against the models clinicians are familiar with like NEWS2
Sensitivity tests on the Medical Event Standard encoding. Do shifts in time cause spurious changes in performance? Do changes in medical acronyms cause spurious changes in performance?
The authors failed to report performance across demographic attributes (i.e. race, sex). See model cards for model reporting by Mitchell et al (2019).

**Relation To Prior Work:**

Yes. The only other work that may be worth discussing is scalable and accurate deep learning for electronic health records by rajkomar et al (2018). While it is evaluated against in GenHPF, it is a prominent proponent of scalable EHRs with a large dataset.

**Summary And Contributions:**

The authors present a dataset MIMIC-Instr and an architecture Llemr to address the heterogenous nature of EHRs for natural language modelling. The GPT-generated questions and answers allow for schema alignment between ClinicalBERT to a decoder only GPT model. The complex case management questions add to the curriculum to improve the complex understanding of the sequence. Both QA and zero shot results were competetive with baselines.

---

> ### Author Rebuttal · Authors · 2024-08-24
>
> We sincerely thank the reviewer for their detailed feedback and the positive evaluation of our work.
>
> ## 1. ClinicalBERT Freezing
>
> We deliberately keep the weights of ClinicalBERT frozen to ensure training efficiency. Training ClinicalBERT alongside LLM is extremely expensive computation-wise. For example, for an input sample with 1000 events, we need to perform 1000 forward passes through ClinicalBERT which dramatically increases the training time and memory cost. In contrast, if we keep the ClinicalBERT frozen, we can pre-compute the event embeddings and load it on the way.
>
> ## 2. Gap Time in Clinical Tasks
>
> We appreciate the suggestion from the reviewer. We wish to emphasize that we have already applied such gap time during pre-processing. Specifically, we filtered events based on the time at which the result was made available (i.e., `storetime` in MIMIC-IV) instead of the time at which the event was charted (i.e., `charttime` in MIMIC-IV). This prevents data leakage for measurements that take extra time to obtain.
>
> ## 3. Additional Results on External Dataset
>
> We appreciate the reviewer's suggestion to further evaluate the model on external datasets. However, the eICU dataset does not contain clinical notes, making it challenging to synthesize QA tasks for clinical reasoning. Therefore, we chose to use the external EHRNoteQA benchmark, as suggested by the reviewer AY1w. This benchmark was initially generated using GPT-4 and subsequently reviewed and refined by three clinicians to ensure clinical relevance. We compared our model, Llemr, with the best-performing baseline, Vicuna-7b-v1.5. The results are presented below.
>
> | Model          | Score              |
> |----------------|--------------------|
> | Vicuna-7b-v1.5 | 52.39 ± 3.14   |
> | Llemr          | 61.79 ± 4.11   |
>
> Our model, Llemr, outperformed the baseline on the unseen EHRNoteQA dataset, demonstrating its robustness and effectiveness in handling diverse clinical tasks.

---

### Official Review · Reviewer_AKvM · 2024-07-22

**Rating:** 7
**Confidence:** 4
**Clarity:** The paper is well written.

**Review:**

* Pros
  * As it is a promising research field to navigate the possibility of utilizing LLMs with EHR data, this work is well motivated and the proposed dataset should be definitely valuable for other researchers working with machine learning for health care.
  * The efficacy of the proposed framework is well demonstrated as regards both for the conversational QA task and the conventional predictive tasks with EHR, in comparison with the vanilla LLMs and the existing foundation models.
* Cons
  * It is not clear how the event encoder in $\textit{Llemr}$ processes each event. From my understanding, after feeding each event, which follows a textual format composed of “{timestamp} {type} {value}”, to the ClinicalBERT, it will have a sequence of embeddings having the same length with the tokenized event. However, in Figure 2 and the main text, it seems that each textual event is mapped to only one embedding token. For each event, do the authors concatenate or aggregate the tokenized embeddings and project them to the word embedding space?
  * It is also unclear how $\textit{Llemr}$ handles the standard clinical predictive benchmark tasks. Did the authors make a specific prompt for each task (Mortality, Readmission, Length-of-Stay, and Diagnosis) and process them to Llemr? If it’s correct, what are those prompts?

**Strengths:**

Please see Pros in the Review section.

**Additional Feedback:**

None.

**Correctness:**

The dataset is constructed in a sound way and the experiments are well formulated.

**Documentation:**

The content of the dataset is well described, but the dataset seems not released at this moment.

**Ethics:**

No.

**Limitations:**

The authors did not include a limitations section for their dataset in the main text, but in the supplementary material.

**Opportunities For Improvement:**

Please see Cons in the Review section.

**Relation To Prior Work:**

Relation to prior work is clearly discussed.

**Summary And Contributions:**

This work proposes a new instruction tuning dataset upon MIMIC-IV tabular EHR data, as well as a framework called MIMIC-Instr to train and evaluate the LLMs with the proposed dataset. The proposed dataset is composed of two types of questions: 1) Schema alignment questions which require the model to extract information from the given EHR event sequences, and 2) Clinical reasoning questions which ask the model to provide clinical reasoning.

---

> ### Author Rebuttal · Authors · 2024-08-24
>
> We thank the reviewer for the valuable feedback and positive evaluation of our work. Below, we address specific concerns from the reviewer.
>
> ## 1. Event Encoding Process
>
> Each event, formatted as “{timestamp} {type} {value}”, is tokenized and passed through ClinicalBERT, generating a sequence of embeddings. We then extract the embedding of the “CLS” token (automatically added by the tokenizer) and use it as the event embedding.
>
> ## 2. Clinical Predictive Benchmark Tasks
>
> We didn’t use any prompts for the clinical predictive tasks as we perform further fine-tuning of the model weight for these tasks.
>
> ## 3. Dataset Release
>
> Due to the user agreements of the MIIMC-IV data, we need to work with PhysioNet in releasing the MIMIC-Instr dataset. We are committed to making it available on PhysioNet upon acceptance.

---

### Official Review · Reviewer_AY1w · 2024-07-28
**Models trained on Med-Instruct perform well on Med-Instruct, but are they better instruction followers on EHR data generally?**

**Rating:** 7
**Confidence:** 4

**Review:**

Quality:
The paper is overall well-written and the experiments are reasonable with strong benchmarks. I have a few concerns about the degree to which the claims made are supported by the results (see “Correctness” below).

Clarity:
I have a few lingering questions after reading the manuscript (see “Clarity” section below), but generally I thought the exposition was clear and compelling. It could benefit from some careful copy editing.

Originality:
I thought the way in which the authors proposed doing schema alignment of event embeddings with the LLM backbone was clever. I’m not familiar enough with the latest and greatest in multimodal learning to appreciate how novel this kind of approach is more broadly, but I haven’t seen it in the EHR foundation model space, at least.

**Strengths:**

Significance of the contribution:
Being able to feed event embeddings into a pre-trained large language model while retaining all the benefits of the hard work that went into pre-training that large language model is an important feat, and it is non-obvious how to do that well. This is an important step in that direction. That being said, this is largely an information compression play that’s only necessary because model context lengths are insufficient. In a world where compute costs trend towards zero and context lengths trend towards infinity, it’s not clear to me yet whether you actually need to do that embedding compression at all. Channeling Richard Sutton’s “bitter lesson”, I’d like to see more compelling evidence in this paper *against* the approach of just serializing everything and not doing any explicit event embedding step. If you’re doing multi-step refinement anyway, then in theory you’re still reading over the entire record. The authors should at least benchmark the best-performing baseline LLM (vicuna-7b-v1.5) on event streams but without the event embedding step (just feed the serialized events into the model). And specifically, the ask is to fine tune this model on the 500k Q/A pairs using this expanded representation and then evaluate (as opposed to Table 3, where my understanding is that it’s evaluation without SFT on the expanded representation).

Relevance to the broader research community:
I think anyone who works on EHR foundation models (not necessarily a massive community, but certainly a vibrant and robust one) will appreciate this paper. Unclear to me whether this is novel enough for folks interested in multimodal learning outside of this space will think this is novel or important (not because I don’t think it is, I’m just not plugged in enough to that line of work).

Quality of the research:
Generally high. I was overall impressed with the scope, ambition, and execution of the work.

Ethical and social implications:
Being able to train better models on EHR data can translate to lives saved and improved efficiency in healthcare. MIMIC-IV is a standard dataset so no concerns about privacy etc. as those have largely all been ironed out.

**Additional Feedback:**

No additional feedback.

**Clarity:**

Is the paper well written? Generally yes, but it could definitely benefit from copy editing to ensure fluid and grammatically correct English. Also tense consistency is an issue here and there (sometimes the paper uses present tense to describe methods/results, sometimes the paper uses past tense). Some examples:
“Previous study shows [...]” on line 21 -> “A previous study showed [...]”
“EHR records” on line 123 -> “EHRs record” or “The EHR records”
“We can see that baselines relied on” on line 292 -> “We can see that baselines relying on” or “We can see that baselines which relied on”
“Specifically, we prompt” on line 183 -> “Specifically, we prompted” (for past tense consistency)
“This demonstrated” on line 299 -> “This demonstrates” (for present tense consistency in the discussion)
Something like Grammarly or Writefull should be able to catch and take care of these.

More broadly, there were a few spots where some additional clarity is warranted. The authors mention on line 237 that “we feed the question, the ground-truth answer, the GPT-4 generated answer, and the candidate LLM generated answer to GPT-3.5”. The way this sentence is constructed it sounds like you’re feeding four things into GPT-3.5. But is what you’re calling “the ground-truth answer” actually just the “GPT-4 generated answer”? If this is the case then I would put “the GPT-4 generated answer” in parentheses rather than separated by a comma because you’re already using a comma here to separate items in the list and so it gets confusing.

Line 61 you say “Another set of 50K QA-pairs were generated from the complementary discharge summaries with GPT-3.5”. What does “complementary” mean here? Complementary to what?

For the 350k schema alignment questions, it seems clear to me that you’re passing into the prompts during training a stream of medical events, each of which is represented by a single embedding based on the serialized description of the event. In the 50k Clinical reasoning subset, though, I understood that the questions are automatically generated from discharge summaries, but during inference are the LLMs given access to the discharge summaries as well? Or are they asked questions that were inspired by the discharge summary but asked to reason over structured clinical events? More broadly, how are notes being incorporated into the event streams?

When generating the reference responses with GPT-4, do you pass in the serialized event streams as is? What about notes? What context length do you use? Do you use multi-step refinement if the patient record doesn’t fit into the model’s context length?

“Llemr” is described at various points in the paper both as a general framework and as a specific model. It would be good to disambiguate the two.

How are digits tokenized? Maybe including an example of how “Hemoglobin 12 g/dL” is tokenized could be helpful. Is it “Hemo”, “globin”, “12”, “g/”, “dL”? Or is it “Hemo”, “globin”, “1”, “2”, “g”, “/”, “dL” for example?

A few of the example questions and answers included “at XXX hours” such as “at 650.05 hour” in line 172. What does this mean? I’m assuming delta time since the start of the patient’s stay? I recognize that dates are fuzzed in MIMIC-IV but they should still be consistent within patient timelines. Why not construct questions that are able to query specific dates? (Common relative time horizons are fine e.g., “within 48 hours of admission” or “within the last 3 days”, but “at 650.05 hour” feels very contrived).

You mention macro AUC-ROC on line 281. Is this one-vs-one or one-vs-rest?

For the “Performance on Standard Clinical Predictive Benchmarks”, does the patient timeline passed to the model incorporate notes? If so, how? (As a single embedding for the document? Concatenation of the token sequence?)

**Correctness:**

I think the authors make a compelling argument based on the results that you can use a Large Language Model backbone and some clever architecture/fine tuning innovations to create a model that effectively reasons over medical event streams. The results on the prediction modeling tasks are hard to argue with, though the bootstrapped standard deviations for those results are all large enough that I do wonder if the gap between Llemr and other methods is statistically significant (the authors should test this explicitly).

One claim that I’m not totally convinced by yet is that better performance on the in-distribution MIMIC-Instr dataset actually translates to better instruction following over EHRs generally. This is because it’s not at all clear to me that the questions generated using templates + GPT-3.5 are actually the kinds of instructions that clinicians would submit to an LLM+EHR system. Ideally, Llemr would be evaluated on one the MedAlign or EHRNoteQA datasets mentioned in Table 1. My understanding is that MedAlign isn’t available as of yet, so EHRNoteQA would probably be the right choice. What got me concerned about question quality is Figure 1b. The question “Which medications should be used for pain control following the surgical procedure?” just doesn’t sound like the kind of question that a practicing clinician would ask conditioned on a patient timeline where the surgery was already performed in the past… Also the example given in line 171 “Which Blood Gas measurement on the Blood specimen were abnormal at the 650.05 hour?” is very strangely worded. I recognize that GPT-3.5’s paraphrase to “Show me the abnormal blood gas measurements at the 650.05 hours” is markedly improved, but also who asks about a time window that specific and what does “at 650.05 hours” even mean?

Also, it was unclear to me if the (instruction, response) pairs in the 50k clinical reasoning set include discharge summaries in the instruction/prompt. If the input prompts do include discharge summaries, then are you really evaluating a model’s ability to reason or are you just training a model to *extract* clinical reasoning that *already exists* within the discharge summary? Then it would seem that SFT on Llemr enables a model to do information extraction over event sequences and (potentially) clinical reasoning over discharge summaries, but it remains unclear if the SFT model can do better clinical reasoning over event sequences and better information extraction over discharge summaries.

Alternatively, if the input prompts do *not* include discharge summaries, how do you know that the input prompt contains sufficient information to answer the clinical reasoning question appropriately? If the input doesn’t contain sufficient information to answer the clinical reasoning question, then how often does the ground truth reference response contain hallucinations?

Is the dataset constructed in a sound way?
Are the evaluation methods and experiment design appropriate and performed correctly? Yes, I appreciated the emphasis both on evaluation of Llemr as a conversational assistant as well as benchmark results for standard clinical predictive tasks. Smart to “outsource” NLG evaluation here to Li et al (leveraging GPT to quantify the quality of generated responses), but I do think having some human evaluation by a subject matter expert could provide more confidence in the evaluation numbers. Various papers such as the Fleming et al (2024) paper you mention showed that even GPT-4 can have high error rates on these kinds of tasks (and we would imagine that GPT-3.5 would have even higher error rates). By evaluating against GPT-4 as the “ground truth” reference response you’d be potentially penalizing a model that performs even better than GPT-4 because the two would sometimes disagree.

**Documentation:**

Is there sufficient detail on data collection and organization?
Generally yes, though hard to evaluate without actually seeing the dataset. See comments on clarity above.

Is there sufficient detail on availability and maintenance?
Yes, in the supplement

Is there sufficient detail on ethical and responsible use?
Yes

Does the dataset have a URL for reviewer to access the dataset?
No, but the authors say that it will be released via PhysioNet.

Does the dataset have a hosting, licensing, and maintenance plan?
Yes

Is there sufficient detail to support reproducibility?
I’d like to see cost estimates:
How much did it cost to generate the dataset in terms of OpenAI API spend?
How much did it cost to evaluate each model?
What GPUs were used for inference and fine-tuning and how much GPU time was required for SFT and evaluation?

**Ethics:**

Are there any ethical concerns with the submission that warrant further review?
Consent: No, standard MIMIC/PhysioNet
Privacy: No, standard MIMIC/PhysioNet
Responsible use: No, standard MIMIC/PhysioNet
Legal compliance: No, standard MIMIC/PhysioNet

**Limitations:**

The authors have adequately addressed the limitations and potential negative societal impact of their work

**Opportunities For Improvement:**

The authors mention the use of multi-step refinement for running inference on EHRs that don’t fit into the models’ context length. What about RAG instead? Or even LangChain’s MapReduceChain? Multi-step refinement tends to yield answers that are overly verbose, in my experience.

If each question template is accompanied with a manually created Python script extracting the ground-truth answer from the corresponding EHR template, can you not pass in the (question template, python script) pairs you have so far as in-context examples to GPT-4 with 128k context lengths and have it generate new question templates and Python scripts for you?

**Relation To Prior Work:**

I think the authors rightly point out that previous contributions have either focused on medicine but not on EHRs (MedQA, MedMCQA, PubMedQA, MMLU) or they have been too small scale (e.g., EHRNoteQA, MedAlign) to support instruction fine-tuning. Given the cost of generating instruction-response pairs in this space (because you need clinicians to manually generate or vet any reference responses) it seems that synthetic data was always going to be the path forward, though how best to generate that data was an open question. This paper provide one compelling answer to that question.

**Summary And Contributions:**

This paper seeks to enable improved instruction-following with Large Language Models over Electronic Health Records by (1) constructing a dataset of 400k synthetic instruction-following data; and (2) introducing “Llemr”, an approach for training Large Language Models on EHR data by aligning event embeddings to a frozen Large Language Model in order to reduce effective context length (because each serialized event is embedded with a single “token”) while maintaining the performance of the underlying pretrained model. The authors mention “Open-source” as a third contribution, namely that they will release the dataset, code, and model weights, but given that MIMIC-IV and many LLMs already “open-source”, this seems more like an expectation given the first two contributions and less of a compelling contribution in its own right. (What’s the use of a dataset and benchmark if nobody else can use them?)

---

> ### Author Rebuttal · Authors · 2024-08-24
>
> We appreciate the detailed feedback and the positive evaluation of our paper. We are pleased that the reviewer found the paper to be well-written and the proposed approach to be significant and novel. Below, we address the specific concerns and suggestions raised by the reviewer.
>
> ## 1. Feed Event Embeddings V.S. Raw Event Text into LLM
>
> We agree with the reviewer that in a hypothetical scenario where compute costs trend toward zero and context lengths trend toward infinity, directly feeding raw event text into LLMs could be a superior approach. This method would eliminate the need for intermediate event embeddings and could potentially preserve more granular details of the patient data.
>
> However, in the current computation setting, the reality is quite different. LLMs are still constrained by finite context lengths and significant compute costs, particularly when dealing with extensive EHR data, where a single patient's record can consist of thousands of discrete events. In such cases, directly feeding raw event text into an LLM would often exceed the model's context length, necessitating complex strategies like truncation or multi-step refinement, which could lead to loss of important information or increased inference times.
>
> Our approach to using event embeddings is designed to mitigate these practical challenges. By compressing each event into a single embedding, we significantly reduce the effective context length, allowing the model to process the entire event sequence in one pass without exceeding its context limitations.
>
> To illustrate the difference, consider the event "5.76 hour, labevents, Blood Iron Chemistry: 90 ug/dL (normal)" If we feed this raw event text directly into an LLM, it would typically be tokenized as: ["5.76", " hour", ",", " labevents", ",", " Blood", " Iron", " Chemistry", ":", " 90", " ug", "/", "dL", " (", "normal", ")"]. Depending on the LLM's tokenizer, this could result in at least 10 tokens. In contrast, Llemr processes this event by first encoding it into a single embedding vector. This difference is critical because it means that Llemr can handle a far greater number of events within the same context window compared to feeding raw text directly into an LLM.
>
> ## 2. On the Use of RAG or LangChain’s MapReduceChain
>
> We appreciate the reviewer's recommendation to consider other approaches like RAG or LangChain’s MapReduceChain. We agree these techniques can potentially address the limitation due to context length. However, the current out-of-box tools are optimized for unstructured text and do not natively support the integration of structured data in a way that preserves the clinical context. For instance, when retrieving relevant information from an EHR, it is not just about finding the most similar text snippet but also understanding the relationships between different types of data, such as how a lab result might inform a treatment decision recorded in the notes. Standard RAG implementations might retrieve related text but could miss how these texts interact over time, which is vital in clinical decision-making. Thus, while RAG and MapReduceChain offer a powerful framework, they require significant customization and fine-tuning to be effective in the EHR domain.
>
> ## 3. Additional Evaluation on EHRNoteQA
>
> We appreciate the reviewer's suggestion to further evaluate the model on external datasets like EHRNoteQA. We have conducted the suggested experiment, comparing the best-performing baseline vicuna-7b-v1.5 and our model Llemr on the EHRNoteQA dataset. The results can be found below.
>
> | Model          | Score              |
> |----------------|--------------------|
> | Vicuna-7b-v1.5 | 52.39 ± 3.14   |
> | Llemr          | 61.79 ± 4.11   |
>
> Our model, Llemr, can still outperform the baseline on the unseen EHRNoteQA dataset. This demonstrates the robustness and effectiveness of our model.
>
> ## 4. Clarification on Whether the Discharge Summary is Feed into the Model
>
> The reviewer inquired whether the input prompts in the 50K clinical reasoning set include discharge summaries and whether this might confound the evaluation of clinical reasoning. We clarify that while discharge summaries are used to generate the questions, they were not included in the input prompts during model training or evaluation. The model was tested on its ability to reason over structured event sequences alone, ensuring that the evaluations truly reflect the model's reasoning capabilities rather than its ability to extract information from the summaries. While we acknowledge that it is possible that the questions generated from discharge summaries may not be answerable by the structured EHR alone, we see this work as a necessary first step in a larger research journey.
>
> ## 5. Clarification on the difference between the “ground-truth answer” and the “GPT-4 generated answer"
>
> The “ground-truth answer” is directly extracted based on the question templates or from the discharge summary. In contrast, the “GPT-4 generated answer” is generated from the EHR event sequence. To assess the quality of the responses, we ask GPT-3.5 to score both the Llemr-generated answer and the GPT-4 generated answer against the ground-truth answer, and report the relative performance of Llemr by calculating the final score as the ratio of the Llemr score to the GPT-4 score.  This allows for a more consistent and reliable comparison between models.
>
> ## 6. Statistical Significance of Results
> We have performed additional statistical tests (paired two-sided t-test) to confirm the significance of our results. The differences between Llemr and the best baseline are statistically significant (p-value < 0.05) for the readmission, length-of-stay, and diagnosis tasks. These results will be included in an updated version of the paper. This approach normalizes the scores and removes the randomness associated with absolute score values, allowing for a more consistent comparison between models.

---

> ### Author Rebuttal · Authors · 2024-08-24
>
> ## 7. Other Minor Questions
>
> > Q: How do you generate reference responses with GPT-4?
>
> A: We directly pass in the raw EHR event sequence. We used the “gpt-4-turbo” model with 128,000  context WINDOW. Multi-step refinement is performed for extremely long inputs.
>
> > Q: How are digits tokenized?
>
> A: We use the BERT tokenizer. The digits are typically tokenized as individual characters.
>
> > Q: Is macro AUC-ROC one-vs-one or one-vs-rest?
>
> A: The macro AUC-ROC is calculated for the diagnosis task which is a multilabel prediction problem. In this context, the macro AUC-ROC is computed using a one-vs-rest approach.

---

### Decision · Program_Chairs · 2024-09-26

**Decision:**

Accept (Spotlight)

**Comment:**

This paper presents Llemr, a novel model for handling Electronic Health Records (EHRs) using large language models (LLMs), and a 400k synthetic dataset based on MIMIC-III and MIMIC-IV for instruction-following tasks. The paper focuses on schema alignment and clinical reasoning, demonstrating how Llemr can be applied to various clinical prediction tasks. Across several reviewer evaluations, the paper is commended for its potential impact, innovative approach, and strong results, though some areas require clarification and further evaluation.

Strengths:
1) Event Embedding Approach: The innovative method of compressing EHR events into embeddings to fit within LLMs’ context lengths is highly praised, showing potential for improving LLM capabilities in complex EHR scenarios.
2) Significant Dataset Contribution: The introduction of a large-scale, high-quality clinical dataset sourced from MIMIC-III and MIMIC-IV is a significant resource for the community, especially for LLM training and evaluation in the healthcare domain.
3) Competitive Results: Llemr outperforms existing baselines on both synthetic and real-world clinical datasets, including EHRNoteQA, with competitive zero-shot performance on in-domain clinical tasks.
4) Broad Community Impact: This paper lays the groundwork for leveraging LLMs in clinical tasks, a field traditionally resistant to LLM applications due to the complex and sensitive nature of EHR data.

Opportunities for Improvement:
1) Clarification of Event Encoding and Schema Alignment: Several reviewers expressed the need for a clearer explanation of how ClinicalBERT encodes events and how embeddings are mapped within the model. More details about the 350k schema alignment data are also necessary.
2) Evaluation of Clinical Tasks: There is some ambiguity about how clinical prediction tasks (e.g., mortality prediction) are evaluated using text-based responses. Detailed explanations and release of the code for reproducibility are strongly encouraged.
3) Comparison Without Event Embedding Compression: A direct comparison between using and not using event embedding compression would help determine the necessity of this step. Similarly, evaluating alternative methods like RAG or LangChain for handling long EHR inputs could be beneficial.
4) Handling Clinical Reasoning Questions: The realism of some clinical reasoning questions was questioned, particularly when the correct answer may not be explicitly in the EHR. More explanation is needed on how correctness is determined, especially for treatment recommendations.
5) Addressing Data Sensitivity and Robustness: Reviewers suggested sensitivity tests for time shifts and medical acronyms, as well as evaluation on more external datasets to verify the model’s generalizability beyond the MIMIC domain.